# *Echinococcus granulosus*-Induced Liver Damage Through Ferroptosis in Rat Model

**DOI:** 10.3390/cells14050328

**Published:** 2025-02-22

**Authors:** Shaohua Zhai, Yueqi Yang, Yang Zhou, Qianqian Lai, Kunlei Li, Songhan Liu, Weilu Li, Feng Gao, Jiyu Guan

**Affiliations:** 1College of Veterinary Medicine, Jilin University, Changchun 130062, China; 18309916937@163.com (S.Z.); gaofeng@jlu.edu.cn (F.G.); 2College of Veterinary Medicine, Xinjiang Agricultural University, Urumqi 830052, China; yangyueqi7788@163.com (Y.Y.); 18324021810@163.com (Y.Z.); lqq369948@163.com (Q.L.); ebh089@outlook.com (K.L.); a807953546@2980.com (S.L.); 17690714100@163.com (W.L.)

**Keywords:** *Echinococcus granulosus*, hepatocytes, ferroptosis, pathological injury

## Abstract

(1) Background: *Cystic echinococcosis* (CE) is an *Echinococcus granulosus*-induced worldwide parasitic zoonosis and is a recognized public health and socio-economic concern. The liver is the major target organ for CE’s infective form protoscolex (PSCs), which causes serious liver damage and endangers the host’s life. Reports show that PSC infection causes liver cell Fe^2+^ metabolism disorder and abnormal deposition of Fe^2+^ in liver cells and results in liver cell death. However, whether PSC-induced liver cell death is associated with ferroptosis remains to be clarified. (2) Methods: Using both an in vivo rat model and an in vitro co-culture of PSCs and the cell system, we studied the histopathological progress of PSCs infection and the cytopathogenesis of PSC-induced cell death in the liver. Hepatic-injury-related ferroptosis signaling pathways were identified by proteomics analysis at various stages of PSCs infection, and physiological and the biochemical indexes and expression of pathway proteins related to hepatic ferroptosis were studied. Ferrostatin-1, a ferroptosis inhibitor, was employed for in vivo interference with early protoscolices infection in rats, and the effects of the inhibition of hepatocyte ferroptosis on hepatocyte injury and the generation of fibrotic cysts were investigated. Additionally, PSCs were exposed to in vitro co-culture with BRL, a rat hepatocyte line, to clarify the direct influences of PSCs on BRL ferroptosis. (3) Results: The results of our in vivo studies revealed that PSCs infection induced Fe^2+^ enrichment in liver cells surrounding the PSCs cysts, cellular oxidation, and liver tissue damage along with the prolongation of PSCs parasitism. The results of our in vitro studies verified the ability of PSCs to directly induce ferroptosis, the formation of fibrotic cysts, and alteration of the iron metabolism of liver cells. The analysis of KEGG signaling pathways revealed that ferroptosis- and ROS-related pathways were significantly induced with PSCs infection. Using Ferrostatin-1 effectively blocked ferroptosis, reversed Fe^2+^ content, reduced liver cell oxidation, and reduced PSC-induced fibrosis cysts. (4) Conclusions: Our study reveals the histopathological progress of PSC infection and the cytopathogenesis of PSC-induced ferroptosis. Ferrostatin-1 effectively blocked PSCs infection and PSC-induced cell death in vivo and in vitro. Accordingly, the inhibition of PSC-induced hepatocyte ferroptosis may be an effective method in the control of *Echinococcus granulosus* infection and should be seriously considered in clinical studies.

## 1. Introduction

*Cystic echinococcosis* (CE) is an *Echinococcus granulosus*-induced worldwide parasitic zoonosis [1,2,3]. According to the estimations made by the foodborne disease burden epidemiology reference group of the World Health Organization, 19,300 lives and 871,000 disability-adjusted life-years are lost globally each year due to CE. The annual costs associated with CE were also estimated to be USD 3 billion because of the treatment of cases and losses in the livestock industry [4]. The prevalence of CE and *Alveolar echinococcosis* (AE) is the highest in northwestern provinces of China (Xinjiang, Xizang, Qinghai, Gansu, Ningxia, Sichuan, Inner Mongolia), and CE is more serious in Xinjiang [5]. The CE infection rates of sheep, cattle, and humans are 4.52%, 4.84%, and 4.0%, respectively, in Xinjiang [6,7]. There is currently no effective treatment for CE disease. Exploring the pathogenic mechanism of CE and liver injury is of great significance for the development of anti-CE disease drugs [8].

The infective form protoscolex (PSC) or hydatidosis occurs at an early larval stage of *Echinococcus granulosus* [9,10]. The liver is the major target organ for CE [11,12], and the parasitism of PSCs can lead to extensive tissue damage and the formation of fibrotic cysts, causing organ dysfunction and endangering the host’s life [13,14]. The formation of fibrotic cysts of PSCs is able to develop a histological barrier between damaged and healthy liver tissues that reduces the host inflammatory response to PSCs and helps PSCs to parasitize in the liver [15,16]. The development of PSCs in the liver alters cell metabolism, differentiation, and function [17,18,19], leading to liver tissue damage and reduced animal performance and growth, posing a serious threat to animal husbandry and farmers [20,21]. Thus, it is important to further understand the effects of PSCs on liver cells in order to reveal the pathogenic mechanism of PSCs in *Echinococcus granulosus*-induced liver damage.

Ferroptosis is a type of iron-dependent programmed cell death [22,23]. Elevated accumulation of Fe^2+^ in cells leads to an intracellular Fenton reaction [24,25], which increases intracellular ROS production, lipid peroxidation, and ferroptosis [26,27]. While the aggravation of liver function damage happens, damaging ROS are heavily produced, and the antioxidant factor GSH is consumed to neutralize damaging ROS [28,29]; however, the excessive ROS causes cell damage and triggers an inflammatory response to release a variety of cytokines and growth factors [30,31], resulting in the activation and transformation of resting hepatic stellate cells into myofibroblasts and then the synthesis of connective tissue proteins, leading to liver fibrosis [32,33]. Reports indicate that PSCs infection can cause liver cell Fe^2+^ metabolism disorder and abnormal deposition of Fe^2+^ in liver cells surrounding cysts [34,35], and infection with *Echinococcus granulosus* results in liver cell death [36,37]. However, whether PSC-induced liver cell death is associated with ferroptosis remains to be clarified.

In this study, we established an in vivo rat model of PSCs infection to clarify the pathogenic role of PSCs in inducing ferroptosis in liver cells. We studied the pathological characteristics, proteomics, and liver cell death in rats at various stages during PSCs infection. We used the ferroptosis inhibitor Ferrostatin-1 in studies of ferroptosis in liver cells. We also studied the direct effect of PSCs on cultured cells to reveal the ability of PSCs to induce ferroptosis.

## 2. Materials and Methods

### 2.1. Declaration of Ethics

All animal tests were approved by the Animal Experiment Ethics Committee of Xinjiang Agricultural University (License: 2022036). All animals were housed in a room with controlled temperature and humidity and a 12 h light/dark cycle, and were fed ad libitum with rat food and purified water in clean rat cages.

### 2.2. Separation and In Vitro Culture of PSCs

Sheep livers with *Echinococcus granulosus* cysts were collected from a local abattoir, and mitochondrial *COX1* gene detection was carried out on the collected PSCs. The cyst solution was extracted and put into a sterilized 50 mL centrifuge tube using a sterile syringe, and the PSCs were deposited at the bottom of the tube by natural sedimentation to remove supernatants. Tissue debris was removed by repeated rinsing with PBS containing streptomycin and penicillin, and the isolated PSCs were cultured in RPMI-1640 containing 10% FBS [38,39].

### 2.3. Establishment of Rat Model of PSC Infection

SD rats (half male and half female, weights = 100 ± 10 g) were purchased from the Experimental Animal Center of Xinjiang Medical University, and were raised for one week for subsequent experiments. The SD rats were anesthetized, and the abdominal cavity was surgically cut to expose the liver. An amount of 0.5 mL of PSCs suspension (20,000/mL) was aspirated and inserted into the right hepatic lobe at a depth of 1 cm at an oblique angle of 45° using a 1 mL syringe. A dry cotton swab was used to stop the bleeding after injection, 0.5 mL of oil procaine penicillin was intraperitoneally injected into the abdominal cavity, and the abdominal wall and skin were sutured in layers, after which the rat was allowed to awaken naturally [40]. For the Ferrostatin-1 inhibitor intervention group, Ferrostatin-1 (concentration 5 mg/kg) was injected intraperitoneally into rats for 21 consecutive days after 7 days of PSCs infection; we repeatedly infected 6 rats in each experimental group [41].

### 2.4. Co-Culture of PSCs and BRL Cells

Pre-cultured BRL cells were digested with trypsin and diluted to 1 × 10^6^/mL. The control group (BRL cells), the BRL cells + PSCs (1000/mL) group [42], and the BRL cells + PSCs (1000/mL) + Ferrostatin-1 (5 μM) group [43] were established. These groups were co-cultured in 10% FBS DMEM medium at 25 cm^2^ for 72 h to monitor the cells’ morphological changes. Additionally, the cells and supernatants were collected for subsequent experiments.

### 2.5. HE Staining

Each rat was anesthetized and executed by cervical dislocation. The liver was extracted, and the rat model liver cyst and surrounding liver tissue (1 × 1 × 0.5 cm^3^) were excised, fixed with 4% paraformaldehyde fixative, dehydrated using a series of gradients of ethanol, treated with xylene, embedded in solid paraffin wax, sliced, and stained with HE staining. The digital information images of the slices were collected by a scanner (Motic, Xiamen, China).

### 2.6. Masson Cytochemical Staining

Paraffin sections of the rat model liver tissue were deparaffinized, rehydrated, and processed using a Masson staining kit (SolarBio, G1340, Beijing, China) for morphological observation of liver cyst fibrous tissue.

### 2.7. Investigation of Ultra-Structure by TEM

The liver tissue around the cyst of the rat model (1 × 1 × 1 mm^3^) was double-fixed with 2.5% glutaraldehyde + 1% osmic acid; BRL cells were scraped and transferred to a 1.5 mL centrifuge tube and centrifuged at 1000 rpm/min for 10 min; the supernatant was aspirated; and the cell precipitates were double-fixed with 2.5% glutaraldehyde + 1% osmic acid. The samples were subjected to dehydration with ethanol, acetone displacement, permeation with Epon 812 resin, embedding, and polymerization (37 °C, 45 °C, and 60 °C for 24 h each), and the samples were sliced by a Leica EM UC7 ultrathin microtome (thickness = 70 nm). Ultrathin sections were double-stained with uranyl acetate–lead citrate, placed in a TEM (120 kV, JEOL-1400, Tokyo, Japan) to observe ultra-structural changes in the cells, and photographed.

### 2.8. Statistics and Analysis of the Number of Inflammatory Cells in Hepatic Cysts

HE sections were selected to image the inflammatory cell area of the liver cyst wall under a microscope with a 40-fold field of view. Three sections were selected in each group, and three fields of view were chosen for each section. After the imaging was completed, Image-Pro Plus 6.0 analysis software was used to unify the millimeter as the standard unit, the number of inflammatory cells in each section was counted, and the corresponding tissue area was measured. The sections were selected to visualize the inflammatory cell area of the liver cyst wall under a microscope with a 40-fold magnification. Three sections were chosen from each group, and three fields of view were examined for each section. After imaging, Image-Pro Plus 6.0 analysis software was utilized to standardize the measurements to millimeters. The number of inflammatory cells in each section was counted, and the corresponding tissue area was measured. The density of inflammatory cells was calculated using the formula density = number of inflammatory cells/tissue area.

### 2.9. Measurement of TGF-β1, TFN-α, and IFN-γ Levels

The cell debris was centrifuged, a total of 50 mg of liver tissue surrounding the cyst was collected, and nine times the volume of Control saline was added to create a tissue homogenate. The mixture was then centrifuged to remove cell debris, and the supernatant was collected as a sample; BRL cells were digested and transferred into a 1.5 mL centrifuge tube. They were centrifuged and rinsed twice with enzyme-free PBS suspension, following the protocols of the Rat TGF-β1 ELISA Kit (Multiscience, EK981-48, Shanghai, China), Rat TNF-α ELISA Kit (Multiscience, 70-EK382/3-48, Shanghai, China), and Rat IFN-γ ELISA Kit (Multiscience, 70-EK380/3-48, Shanghai, China), and the supernatant was collected as a sample. The BRL cells were digested, transferred into a 1.5 mL centrifuge tube, centrifuged, and rinsed twice with enzyme-free PBS suspension according to the Rat TGF-β1 ELISA Kit (Multiscience, Ek981-48, Shanghai, China); Rat TNF-α ELISA Kit (Multisciences, 70-EK382/3-48, Shanghai, China); and rat IFN-γ ELISA Kit (Multiscience, 70-EK380/3-48, Shanghai, China).

### 2.10. Proteomics DIA Sequencing Method

A total of 200 mg/rat of liver tissue was collected around the rat model cysts, the liver tissue samples were added into a grinding tube, 2 mL of protein extraction buffer was added and mixed, and the proteins were ground thoroughly for extraction. The total protein extracted from the samples was quantified using the BCA method. A total of 100 µg of protein was removed from each of the above samples, and protein enzymatic hydrolysis was performed using the FASP method. For peptide samples after enzymatic hydrolysis, equal amounts of each biological sample were taken and mixed, and 100 µg was taken for high-pH reversed-phase (RP) chromatography classification. The DDA data collected after grading were subjected to protein retrieval analysis by Proteome Discoverer 2.1.0182 (Thermo Fisher Scientific, Rockford, IL, USA). DIA data were collected separately from the biological samples and quantitatively analyzed using Skyline (Department of Genome Sciences, University of Washington, Ave. NE, Seattle, WA). Hierarchical cluster analysis was performed using both quantified proteins and samples, and all identified proteins were functionally annotated using the GO, KEGG, egg NOG, Pfam, and SignaIP databases.

### 2.11. Detection of ROS by Flow Cytometry

A total of 50 mg of liver tissue around rat liver cysts was collected and rinsed with PBS to remove blood cells; the liver tissue was ground with a 200-mesh sieve, rinsed with PBS, and centrifuged to collect the single-cell suspension, and the cells were diluted to 1 × 10^7^/mL. A total of 500 μL of cell suspension was added into to the sample tube, DCFH-DA with a final concentration of 10 μM was added, and the sample was incubated at 37 °C for 30 min; the sample was rinsed twice with pre-cooled PBS, 500 μL of PBS was added, the solution was transferred to a flow tube, and the fluorescence intensity was measured by FITC channel flow cytometry.

### 2.12. Measurement of Fe^2+^, LDH, MDA, SOD, and GSH Levels

A total of 50 mg of liver tissue was collected around the cyst, 9 times the normal saline was added to make a tissue homogenate and centrifuged to precipitate cell debris, and the supernatant was collected; the BRL cells were digested, transferred into a 1.5 mL centrifuge tube to precipitate, and rinsed twice with enzyme-free PBS suspension to collect cells for detection. An Fe^2+^ kit (SolarBio, BC5415, Shanghai, China), an LDH kit (Jiancheng Bioengineering Institute, A020-2, Nanjing, China), a lipid peroxidation (MDA) kit (Jiancheng Bioengineering Institute, A003-1,Nanjing, China), a SOD kit (Jiancheng Bioengineering Institute, A001-3, Nanjing, China), and a total glutathione/oxidative glutathione (T-GSH/GSSG) kit (Jiancheng Bioengineering Institute, A061-1, Nanjing, China) were used.

### 2.13. CCK-8 Test

The BRL cells were digested and diluted to 1 × 10^6^/mL to prepare a cell suspension. A total of 200 μL of cell suspension was added to 24-well cell migration plates, cell culture chambers were placed in the wells [44], and 200 μL of PSC suspension (1000/mL) was added to the cell culture chamber and incubated in a 37 °C incubator. The cell culture chambers were removed for different incubation times (24 h, 48 h, 72 h), and 10% CCK-8 reagent (Servicebio, G4103, Wuhan, China) was added to the cell culture plate wells, and the cells were left to incubate for 1 h at 37 °C. The OD value of the culture solution of each cell well was measured using a wavelength of 450 nm, and the cell inhibition rates were calculated as follows: inhibition rate % = [1 − As/Ac] × 100%; here, AS is the OD of the experimental well, and Ac is the OD of the control well.

### 2.14. Western Blotting

A total of 50 mg of liver tissue was collected around the cysts. Liver tissue and cellular proteins were extracted using RIPA lysis buffer, and protein content was determined using a BSA protein assay kit (TransGen Biotech, DQ111-01, Beijing, China). The protein samples were diluted to the same concentration with RIPA lysis buffer, and 1/4 of the volume of the SDS-PAGE sample was added to load buffer and heated at 100 °C for 10 min. A total of 50 μg of protein samples were subjected to SDS-PAGE gel electrophoresis, and then transferred to PVDF membranes according to the standard procedure. The membrane was sealed with skim milk dissolved in TBST for 2 h. The primary antibody was incubated with the PVDF membrane at 4 °C overnight, and the secondary antibody was incubated at room temperature for 1.5 h. Protein bands were observed and photographed using an enhanced chemiluminescence detection system (BioRad, Hercules, CA, USA).

### 2.15. Antibodies

The antibodies used in this study were as follows: TFRC (Boster, 1: 500, Rabbit, BA0462-2, Wuhan, China); Caspase-3 (Boster, 1:500, Rabbit, BA3257, Wuhan, China); GSDMD (Proteintech Group, 1:800, Rabbit, 20770-1-AP, Wuhan, China); LC3I/II (Abcam, 1:1000, Rabbit, ab192890, Shanghai, China); Anti-GPX4 Antibody (Boster, 1:400, Rabbit, BM5231,Wuhan, China); Anti-FTH1 Antibody (Boster, China, 1:400, Rabbit, BM4487,Wuhan, China); Anti-NOX1 Antibody (Boster, China, 1:400, Rabbit, BA3720,Wuhan, China); Anti-CD98 Antibody (SLC3A2, Boster, China, 1:400, Rabbit, A01794-1,Wuhan,China); Anti-xCT Antibody (SLC7A11,Abcam,1:400, Rabbit, A01794-1, Shanghai, China); β-actin (Sino Biological, 1:1000, Mouse, 100166-MM10, Beijing, China).

### 2.16. Statistical Analysis

All images were created in Adobe Illustrator (2024) and Adobe Photoshop software (2021). Statistical analysis and mapping were performed using GraphPad Prism 9.5.0 according to the experimental type, and a double-tailed Student’s *t* test or a one-way ANOVA was performed to analyze the significance of the differences. The *p* values are displayed using the GP format (**** *p* < 0.0001, *** *p* < 0.001, ** *p* < 0.01, * *p* < 0.05, ns *p* ≥ 0.05), and all graphical data are expressed as the mean ± SD.

## 3. Results

### 3.1. Pathological Characteristics of Rat Liver Tissue Injury at Different Stages of Infection Caused by PSCs

To establish an in vivo rat model of PSCs infection, we injected PSCs into the right hepatic lobe of rats and studied the formation of fibrotic cysts in the liver after injection for 1, 3, and 6 months (Figure 1A and Appendix A). As shown in Figure 1B(a), the pathological examinations detected that the surface of healthy liver was smooth and flat. In contrast, bean-sized, gray-white cyst bulges developed by 1 month after the infection (Figure 1B(b)), large gray-white vesicles developed with tense walls and full of cystic fluid by 3 months (Figure 1B(c)), and multiple cysts with transparent walls and filled with fluid developed by 6 months (Figure 1B(d)). These results indicated that the degrees of liver damage were increasingly associated with the volumes of fibrotic cysts after PSCs infection.

The histological studies of liver tissues with cysts revealed that the central vein, hepatic cord, and hepatic sinusoids of healthy lobules were well organized (Figure 1C(a,b) and Appendix A). In contrast, PSCs were detectable in the liver parenchyma, hepatocytes were reduced at the parasitic sites, and inflammatory cells were populated around the lesions after infection for 1 month (Figure 1C(c,d) and Appendix A). After 3 months of infection, the size of the cysts increased, the walls of fibrotic cysts thickened, and the boundary between cysts and peripheral tissue became clear (Figure 1C(e,f) and Appendix A). By 6 months, the cyst volume had increased, the walls of fibrotic cysts had become thinner, and the peripheral inflammatory cells had reduced (Figure 1C(g,h) and Appendix A). These results indicated that PSCs caused tissue damage associated with the development of cysts in the liver.

The statistical analysis of inflammatory cells detected in the peripheral tissues of cysts revealed that the number of inflammatory cells had significantly increased (~4-fold) by 1 month and remained (~3-fold) 3 and 6 months after PSCs infection (Figure 1D and Appendix A). The studies of cytokines in the hepatocytes adjacent to cysts revealed increases in TGF-β1 (~4-fold), TNF (~7.5-fold), and IFN (~6-fold) along with PSCs infection (Figure 1E–G). These results indicated that the inflammatory reaction was associated with cyst development and the degree of liver tissue damage.

To study cell death in the peripheral tissue of cysts, we determined the level of cell death markers (TFRC, Caspase-3, GSDMD, LC3I/II), and we detected that the TFRC and Caspase-3 levels increased, but not GSDMD or LC3I/II, along with PSCs infection (Figure 1H(a,b)), indicating that liver tissue damage was associated with cell death, possibly due to apoptosis or ferroptosis.

### 3.2. Proteomics Detection of Rat Peripheral Liver Tissue at Different Stages of Infection with PSC Cysts

To study the changes in protein expression and cellular pathways induced by PSCs infection, we isolated peri-cyst hepatocytes for proteomic analysis. As shown in Figure 2A, when compared with the control group, ~2993 and ~2613 protein expression was up- and down-regulated, respectively, with after 1 month of infection (A), ~2859 and ~2747 protein expression was up- and down-regulated, respectively, after 3 months of infection (B), and ~3000 and ~2606 protein expression was up- and down-regulated, respectively, after 6 months of infection (C). KEGG pathway analysis revealed that 79, 85, and 56 differentially PSC-enriched proteins (Figure 2D–F) were primarily associated with ferroptosis-related pathways, enriched with significantly differentially expressed proteins FTH1 and FTH1 associated with the Ferroptosis pathway FRIL1 (Appendix A), indicating that the death of hepatocytes was possibly due to ferroptosis induced by intrahepatic parasitism of cysticercosis in vivo.

### 3.3. Detection of Hepatocyte Ferroptosis-Related Indicators of Rats at Different Stages of Infection with PSCs

To determine whether ferroptosis was involved in hepatocercoid parasitism, we isolated liver cells adjacent to cysts from rats for the following studies. When studying the ultra-structure of hepatocytes, we observed that they had a mitochondrial morphology with a defined spinal membrane of healthy hepatocytes (Figure 3A(a,b)). After PSCs infection, the hepatocytes showed enlarged and increased mitochondria with thickened membranes, and parasitic cysts in the hepatic tissue, as well as dense cyst walls and PSCs in the inner wall of the cyst by 1 month, in addition to a disrupted membrane structure, organelle spillage, and vacuolar changes (Figure 3A(c,d)). By 3 months, we detected that the mitochondria density of the spinal electron cloud had deepened, and the volume had reduced (Figure 3A(e,f)). We detected altered liver cell membranes, overflowing organelles, and increased vacuoles by 6 months (Figure 3A(g,h)).When studying biochemical properties, we detected increased Fe^2+^ concentration (>2-fold) (Figure 3B), decreased glutathione (GSH) (Figure 3C), increased intracellular ROS (>2-fold) (Figure 3D), MDA (>4-fold) (Figure 3E), decreased intracellular SOD (Figure 3F), and increased LDH (~2-fold) (Figure 3G) along with the PSCs infection. We also detected increased intracellular TFRC and NOX1, as well as decreased intracellular FTH1, intracellular GPX4, SLC3A2, and SLC7A11 (Figure 3H(a,b)). These results indicated that PSCs infection of the liver resulted in cytostructural alterations, elevated Fe^2+^ content, induced ferroptosis-related and ROS-associated pathways, and reduced antioxidant enzymes in vivo.

### 3.4. Ferrostatin-1 Interferes with PSC-Induced Hepatocyte Ferroptosis

To determine whether ferroptosis played an essential role in liver cell death induced by PSCs, we injected the ferroptosis inhibitor Ferrostatin-1 intraperitoneally into rats infected with PSCs (Figure 4A). As shown in Figure 4B(a)–(c), Ferrostatin-1 treatment reduced PSC-induced hepatocyte ferroptosis, cyst volume on the liver surface, PSCs, and inflammatory cells, indicating a reduced degree of hepatic injury (Figure 4C and Appendix A). The TEM study revealed that the nuclear morphology of Ferrostatin-1-treated hepatocytes was regular, the endoplasmic reticulum membrane was visible, and the mitochondrial morphology was regular and uniformly distributed (Figure 4D). Ferrostatin-1 treatment also significantly reduced the Fe^2+^ (>2-fold) concentration in hepatocytes (Figure 4E), decreased intracellular ROS (Figure 4G(a,b)), and restored intracellular GSH (>2-fold) (Figure 4F) and SOD (Figure 4H) concentration from PSCs infection. In addition, Ferrostatin-1 treatment reduced the hepatic injury indicators LDH (~2.5-fold) and MDA (>2-fold) (Figure 4I,J) as well as the PSC-induced ferroptosis indicator TFRC (Figure 4K(a,b)). On the other hand, Ferrostatin-1 treatment induced ferroptosis regulators, such as GPX4, FTH1, and SLC7A11 (Figure 4K(a,b)). Accordingly, in vivo Ferrostatin-1 treatment blocked the PSC-induced ferroptosis in liver cells, hepatocyte injury, and hepatic fibrosis.

### 3.5. Effects of Co-Culture of PSCs and BRL Cells on Cell Proliferation and Lipid Peroxidation Level

To verify the effects of PSCs on cultured cells, we used a co-culture model (Figure 5A). As shown in Figure 5B, BRL cells co-cultured with PSCs resulted in a reduced growth rate and cell atrophy that was blocked by Ferrostatin-1. The TEM study revealed that PSC-cocultured BRL cells showed an increased number and volume of mitochondria with an elliptical shape and a thickened membrane that were affected by Ferrostatin-1 treatment (Figure 5C). In addition, the CCk-8 assay revealed that PSCs inhibited BRL cell proliferation (Figure 5D) and increased the lipid peroxidation of ROS that were alleviated by Ferrostatin-1 treatment (Figure 5E(a,b)). These results indicated that PSCs were able to interact directly with hepatocytes and induce cellular alterations involving ferroptosis.

### 3.6. Detection of Relevant Parameters After Co-Culture of PSCs and BRL Cell Ferroptosis

To further verify the involvement of ferroptosis in PSC-induced cell death, we used Ferrostatin-1 to block the ferroptosis pathway induced in BRL cells by PSCs. As shown in Figure 6, Ferrostatin-1 treatment inhibited PSC-increased Fe^2+^ (Figure 6A) and PSC-reduced GSH content (Figure 6B), and increased the GSSH level (Figure 6C), PSC-reduced GSH/GSSH ratio (Figure 6D), PSC-reduced SOD (Figure 6E), and PSC-induced liver damage indexed by LDH and MDA (Figure 6 F,G). The immunoblotting assays showed that all the PSC-induced changes in TFRC, NOX1, GPX4, FTH1, SLC3A2, and SLC7A11 were reduced by Ferrostatin-1 treatment (Figure 6H(a,b)). Accordingly, these results verified that Ferrostatin-1 was effective in inhibiting PSC-induced ferroptosis-related pathways, indicating that this was the key role ferroptosis played in PSC infection in liver cells.

## 4. Discussion

In this communication, we reported, for the first time, that intrahepatic parasitism of *Echinococcus granulosus* resulted in ferroptosis in liver cells, contributing to PSC-induced damage to liver cells. Our in vivo studies revealed that PSCs infection resulted in Fe^2+^ enrichment in liver cells surrounding PSCs cysts, increased cellular oxidation, and liver tissue damage [45,46]. Liver tissue damage and fibrotic cyst volume were increased with the prolongation of PSCs parasitism. Administering an inhibitor blocked ferroptosis and reversed Fe^2+^ content, reduced liver cell oxidation, and reduced PSC-induced fibrosis cysts in the livers of the animals. Our in vitro studies verified the ability of PSCs to directly induce ferroptosis in liver cells, the formation of fibrotic cysts, and the alteration of iron metabolism.

Different degrees of iron metabolism disorders and lipid peroxide accumulation have been found in various liver diseases characterized by ferroptosis. In a rat model of diabetes induced by streptozotocin (STZ), the levels of GSH and SOD in liver tissue decreased, while the level of MDA increased. The increased expression of TFR1, FTH, and FTL indicates that liver injury in diabetes patients may be caused by iron death [47]. The circadian transcription factor ARNTL is a key regulatory factor expressed in various antioxidant or membrane repair systems, such as SLC7A11, GPX4, SOD1, TXN, NFE2L2, and CHMP5, to inhibit iron-induced tissue damage. ARNTL prevents experimental acute pancreatitis by blocking the release of HMGB1 mediated by ferroptosis, revealing a new link between the circadian clock and ferritin response in inflammation and pancreatic injury [48,49,50]. In the detection of Toxoplasma gondii infection in brain cells, it was found that the expression of Fe^2+^ metabolism disorder TFRC was up-regulated. At the same time, SLA711 and GPX4 were down-regulated, indicating that Toxoplasma gondii infection can cause iron death in brain cells and lead to brain damage [51]. Aslam H et al. evaluated the significant increase in GST and GPX levels, ROS production, MDA, and protein carbonyl (PC) levels in the livers of water buffalo naturally infected with cystic echinococcosis. Enhanced MDA damages the cell membrane, releasing the liver injury markers AST, ALT, ACP, and ALP [52]. Wen et al. study the in vivo and in vitro anti-CE effects of artemether (AS). AS can significantly improve liver biochemical indicators in infected mice, reduce TNF-α levels, decrease serum GSH/GSSG ratios, reduce H_2_O_2_ levels in hydatid fluid, and significantly increase T-SOD levels [53].

The results of using an animal PSCs infection model and analyzing the KEGG cell signaling pathway in liver cells revealed that ferroptosis-related signaling pathways (P3VP0, P6VP0, P3VP1; Appendix A) were significantly enriched with PSC infection. PSCs infection induced changes in cell metabolism and ROS-related pathways in association with ferroptosis, contributing to liver tissue damage [54,55]. It is possible that the PSC-induced metabolic imbalance in liver cells was due to the uptake of nutrients from surrounding tissues [56,57], resulting in ferroptosis in liver cells. In addition, the administration of the inhibitor Ferrostatin-1 into *Echinococcus granulosus*-infected rats was able to decrease PSC-increased tissue fibrosis, cyst volume, inflammation, Fe^2+^, and lipid peroxide, as well as ferroptosis-related ROS, GSH, NOX1, SOD, TFRC, GPX4, FTH1, SLC3A2, and SLC7A11 in the liver. Interestingly, Ferrostatin-1 treatment not only intervened with hepatocyte ferroptosis [58,59], but also blocked PSC development in the liver. Whether hepatocyte ferroptosis contributes to PSC development in the liver remains to be determined.

Our initial study using an in vivo animal model, and subsequent study using an in vitro co-culture system, enabled us to determine the histopathological progress of PSCs infection and cytopathogenesis of PSC-induced ferroptosis. Using the inhibitor Ferrostatin-1 in vivo and in vitro effectively blocked PSCs infection and PSC-induced liver cell death. However, whether the secretion–excreta of secretory–excretory products can promote hepatocyte ferroptosis during intrahepatic parasitism and the pathogenic mechanism of PSCs in hepatocyte ferroptosis need to be clarified. 

## 5. Conclusions

Our research revealed the ability of PSCs to induce ferroptosis in liver cells, cellular oxidation, and liver tissue damage along with the prolongation of PSCs parasitism. The inhibition of PSC-induced hepatocyte ferroptosis may be an effective method in the control of *Echinococcosis* and should be seriously considered in clinical studies.

## Figures and Tables

**Figure 1 cells-14-00328-f001:**
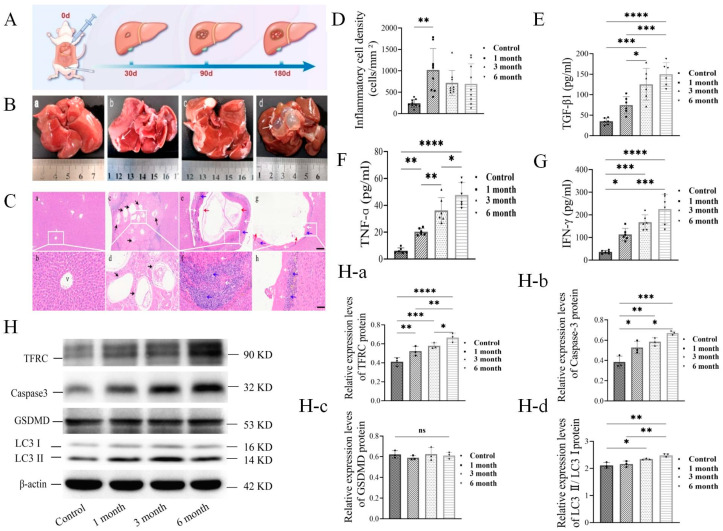
(**A**) Establishment of rat infection model by PSCs liver lobe injection. (**B**) Morphological characteristics of rat liver cysts 1, 3, and 6 months after infection with PSCs. Cysts are indicated by white arrows (*n* = 6 rats/group). (**C**) Histopathological observation results of rat liver cysts at different stages of infection with PSCs; germinal layer, keratinization layer, and inflammatory cell zone are indicated by black, red, white, and blue arrows (*n* = 6 rats/group, scale bars represent 100 μm and 50 μm). (**D**) Statistical results of inflammatory cells in rat liver cysts in peripheral liver tissue at different stages of infection with PSCs (*n* = 3 rats/group). (**E**–**G**) Inflammatory cytokine contents of TGF-β1, TNF, and IFN in rat liver cysts in peripheral liver tissue at different stages of infection with PSCs (*n* = 3 rats/group). (**H**) Western-blotting detection of expression level of death marker proteins ((**H**(**a**)–(**d**)) Ferroptosis TFRC, Apoptotic caspase3, Autophagy GSDMD, Pyroptosis LC3 II/I) in rat hepatocytes at different stages of infection with PSCs (*n* = 3 rats/group); * *p* < 0.05, ** *p* < 0.01, *** *p* < 0.001, **** *p* < 0.0001 (Student *t* test).

**Figure 2 cells-14-00328-f002:**
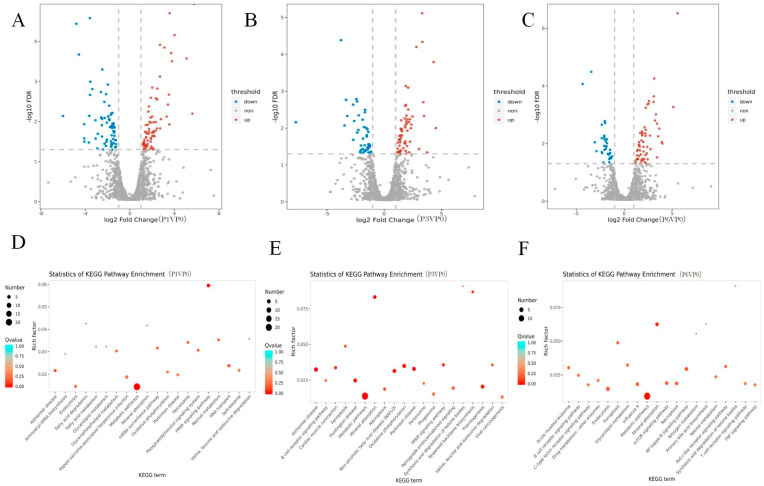
(**A**–**C**) Rat liver differential protein statistical results at different stages of infection with PSCs; (**D**–**F**) rat liver differential KEGG signaling pathway at different stages of infection with PSCs.

**Figure 3 cells-14-00328-f003:**
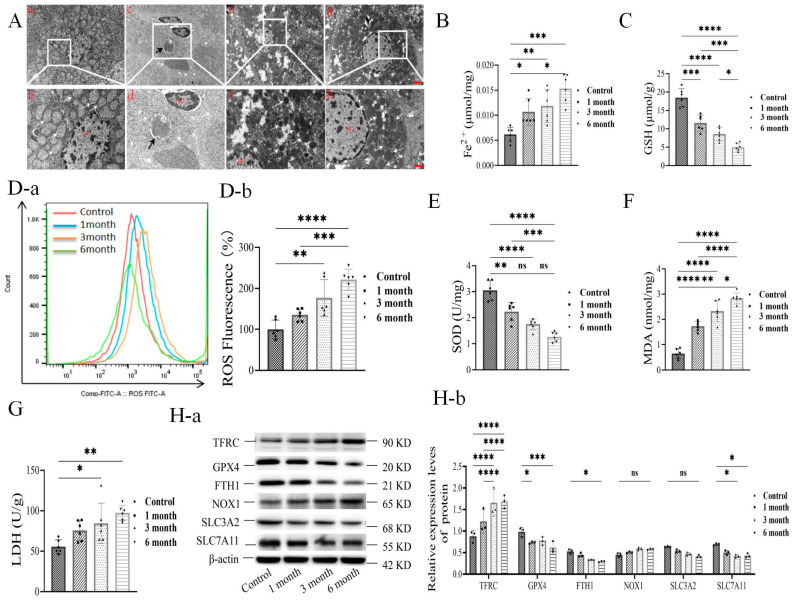
**(A)** Ultra-structure of hepatocytes near cysts at different stages of infection with PSCs (0, 1, 3, and 6 months); mitochondria are indicated by white arrows,PSC are indicated by black arrows(*n* = 3 rats/group, scale bars represent 2 μm and 1 μm). (**B**) Fe^2+^, (**C**) GSH, (**D**(**a**,**b**)) ROS, (**E**) MDA, (**F**) SOD, and (**G**) LDH of hepatocytes near cysts at different stages of infection with PSCs. (**H**(**a**,**b**)) Western blotting of protein expression levels of TFRC, GPX4, FTH1, NOX1, SLC3A2, and SLC7A11 in ferroptosis signaling pathway in rat liver tissue at different stages of infection with PSCs (*n* = 3 rats/group)); * *p* < 0.05, ** *p* < 0.01, *** *p* < 0.001, **** *p* < 0.0001 (Student *t* test).

**Figure 4 cells-14-00328-f004:**
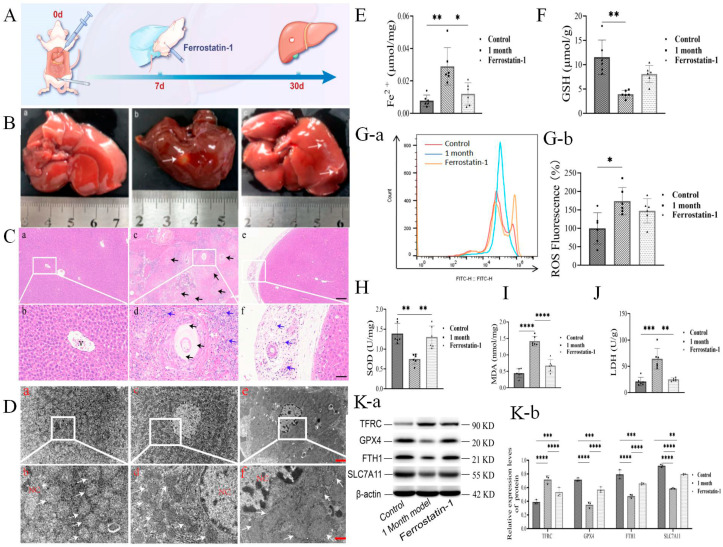
**(A)** Ferrostatin-1 inhibitor intervenes in vivo in PSC-mediated hepatocyte ferroptosis model establishment. (**B**) Characteristics of rat liver cyst morphology in rat blank control group, 1-month infection model group, and Ferrostatin-1 (5 mg/kg) intervention group. Cysts are indicated by white arrows (*n* = 6 rats/group). (**C**) Histological changes in Ferrostatin-1 on liver cysts in a rat model infected with PSCs for 1 month; germinal layer, keratinization layer, and inflammatory cell zone are indicated by black, white, and blue arrows (*n* = 6 rats/group; scale bars represent 100 μm and 50 μm). (**D**) TEM observation of Ferrostatin-1 on cyst peripheral hepatocyte ultra-structure changes in a rat model infected with PSCs for 1 month; mitochondria are indicated by white arrows (*n* = 3 rats/group; scale bars represent 2 μm and 1 μm). Effect of Ferrostatin-1 on (**E**) Fe^2+^, (**F**) GSH, (**G**(**a**,**b**)) ROS, (**H**) SOD, (**I**) LDH, and (**J**) MDA in cyst peripheral liver tissue in rat model infected with PSCs for 1 month; (**K**(**a**,**b**)) Western-blotting assay of Ferrostatin-1 on expression levels of ferroptosis signaling pathway proteins, such as TFRC, GPX4, FTH1, and SLC7A11, in liver tissues in rat model infected with PSCs for 1 month (*n* = 3 rats/group); * *p* < 0.05, ** *p* < 0.01, *** *p* < 0.001, **** *p* < 0.0001 (Student *t* test).

**Figure 5 cells-14-00328-f005:**
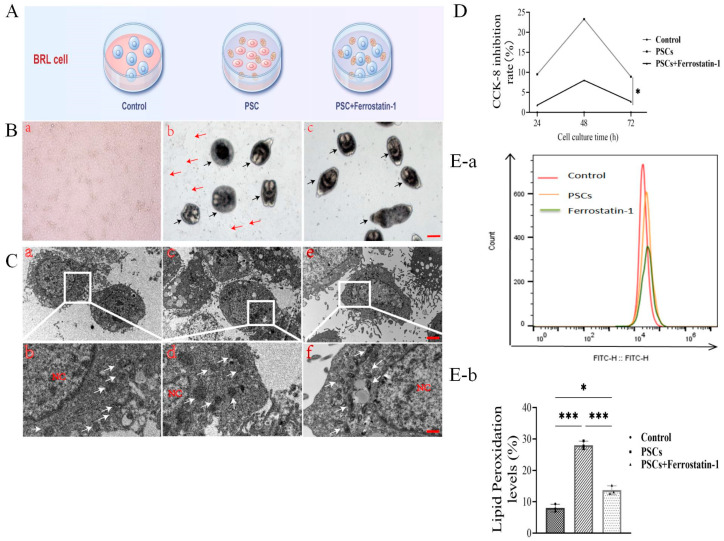
(**A**) Normal BRL cell, PSCs + BRL cell, PSCs+ Ferrostatin-1 + BRL cell, and cell infection models were set up. (**B**) TEM observation of morphology of PSCs and BRL co-cultured cells. Shedding cell plaques are indicated by red arrows, and PSCs are indicated by black arrows (*n* = 6 in every group; scale bars represent 100 μm). (**C**) TEM observation of ultra-structure changes in PSCs and BRL co-cultured cells. Mitochondria are indicated by white arrows (*n* = 3 rats/group; scale bars represent 2 μm and 1 μm). (**D**) CCK-8 detection of effect of PSCs infection on BRL cell proliferation. (**E**(**a**,**b**)) ROS detection of co-culture of PSCs and BRL cells (*n* = 3 rats/group); * *p* < 0.05, *** *p* < 0.001 (Student *t* test).

**Figure 6 cells-14-00328-f006:**
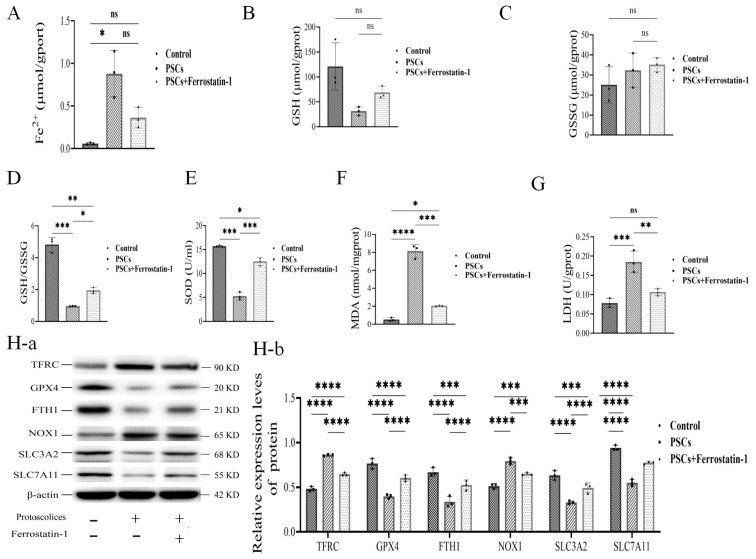
(**A**) Fe^2+^ in normal BRL cells, PSCs and BRL co-cultured cells, and PSCs +Ferrostatin-1 and BRL co-cultured cells. (**B**) GSH, (**C**) GSSH, (**D**) GSH/GSSH, (**E**) SOD, (**F**) LDH, and (**G**) MDA concentration detection (*n* = 6 rats/group). (**H**(**a**,**b**)) Western-blotting detection of expression levels of cell ferroptosis signaling pathway proteins in each group, such as TFRC, GPX4, FTH1, NOX1, SLC3A2, andSLC7A11 (*n* = 3 rats/group); * *p* < 0.05, ** *p* < 0.01, *** *p* < 0.001, **** *p* < 0.0001 (Student *t* test).

## Data Availability

All procedures contributing to this work comply with the ethical standards of the relevant national and institutional guides on the care and use of laboratory animals. Emails requesting details of the materials may be sent to Shao-Hua Zhai, Yue-Qi Yang, or Yang Zhou.

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
