# Peer review of "Echinococcus granulosus-Induced Liver Damage Through Ferroptosis in Rat Model"

_cells, 2025, doi:10.3390/cells14050328_

Round 1

Reviewer 1 Report

Comments and Suggestions for Authors

Manuscript 3442909 by Zhai et al. describes the influence of Echinococcus granulosus in an in vivo liver infection model as well as in an in vitro model. The following aspects deserve additional attention by the authors in order to further improve the manuscript.

-          Text: throughout the text, blank spaces between words are missing regularly, most likely due to formatting errors. Please check the entire text.

-          In Fig. 1C, the authors should provide additional information on the set of criteria employed for the definition of germinal layers, keratinization layers, and inflammatory zones.

-          In Fig. 1 D-H, the authors should provide more information on how the identification of inflammatory cells as well as the analysis of cytokines and protein expression was performed. Precise information on the tissue employed for analysis (e.g. distance from cyst, volume etc.) is required. It should also be clarified how hepatocytes and other cell types were isolated and separated from the tissue.

-          Fig. 2: KEGG pathway analysis. The authors should provide more information on the meaning of the color-code, and on the diameters in graphs D-F. It is also not apparent why the authors focused on ferroptosis, based on the experimental data shown. This aspect needs further clarification. It is also important to consider that ferroptosis, in contrast to e.g. apoptosis, is not characterized by classical activation pathways. Therefore, the individual targets summarized under the term “ferroptosis” in the KEGG analysis should be indicated in detail so that the reader gets a better impression on the targets modified in their expression.

-          In Fig. 1H, the authors illustrated elevated caspase-3 expression, which is a classical hallmark of apoptosis and consequently in contrast to the assumptions made in the present manuscript. It could be speculated that a sequential or mixed contribution of both ferroptosis and apoptosis might be involved.

-          Chapter 3.3. It is not apparent how the effects observed in the images in Fig. 3A are related to ferroptosis. Magnification of the images seems to be not identical. So, the conclusion that mitochondria would show differences in size appear as rather vague.

-          The authors need to explain the presence of intracellular cysts. PSC is way larger than an individual hepatocyte, how can it then be present within a cell.

-          Fig. 3C: it should clarified whether LDH was measured within the cell or in the supernatant.

-          Fig. 4 in the manuscript files does not match the text. Correct Fig. 4 is available to the reviewer.

-          Ferrostatin, not Ferrostain

-          Fig. 6A: the graph indicates iron levels close to zero in the controls. This is impossible, usually intracellular total iron levels can vary between ca. 50 % and 200 % of control.

-          Fig. 6 B+C+D: the GSH/GSSG ratio is rather unusual. In healthy, resting cells, GSSG levels are in the range of 1-5 %. The data illustrated indicate either massive oxidative stress under resting conditions, or difficulties in the experimental detection of GSH and GSSG.

-          Fig. 6G: please indicate whether LDH was detected in the cells or in the supernatant.

-          The authors observed an inhibitory influence of Ferrostatin-1 on cyst growth. This however represents an alternative concept to the direct inhibitory influence on hepatocytes. The authors report that Ferrostatin-1 prevents iron accumulation. The compound however acts as antioxidant, preventing from lipid peroxidation. This indicates that the size of the cyst, respectively load with PSC might directly correlate with hepatocyte damage, potentially by factors from the parasite that evoke types of cell death distinct from ferroptosis. Therefore, the authors should investigate whether the size of the cysts correlate with the changes detected in hepatocytes. As Ferrostatin-1 apparently inhibits cyst growth, the concept of ferroptosis as main driver of hepatocytes cell death is severely challenged and should be verified carefully.

Comments on the Quality of English Language

throughout the text, blank spaces between words are missing regularly, most likely due to formatting errors. Please check the entire text.

Language editing is recommended

Author Response

Responses to comments and suggestions to Reviewer #1

General comments:Text: throughout the text, blank spaces between words are missing regularly, most likely due to formatting errors. Please check the entire text.

Response:The errors in language expression, grammar and spelling have been corrected.

General comments:

In Fig. 1C, the authors should provide additional information on the set of criteria employed for the definition of germinal layers, keratinization layers, and inflammatory zones.

Response:In Fig. 1C, the labeling of the germinal layer of cysts was modified (red arrow). A clear distinction is made between PSCs (black arrows), germinal layers (red arrows), keratinization layers (white arrows), and inflammatory zones (blue arrows), as illustrated in Figure Legen.

General comments:

In Fig. 1 D-H, the authors should provide more information on how the identification of inflammatory cells as well as the analysis of cytokines and protein expression was performed. Precise information on the tissue employed for analysis (e.g. distance from cyst, volume etc.) is required. It should also be clarified how hepatocytes and other cell types were isolated and separated from the tissue.

Response:We added "2.8 Statistics and analysis of the number of inflammatory cells in hepatic" to "2. Materials and Methods" cysts "2.9 Measurement of TGF-β1, TFN-α and IFN-γ levels", and "2.12. Western-blotting" protein sample collection amount.

General comments:

KEGG pathway analysis. The authors should provide more information on the meaning of the color-code, and on the diameters in graphs D-F. It is also not apparent why the authors focused on ferroptosis, based on the experimental data shown. This aspect needs further clarification. It is also important to consider that ferroptosis, in contrast to e.g. apoptosis, is not characterized by classical activation pathways. Therefore, the individual targets summarized under the term “ferroptosis” in the KEGG analysis should be indicated in detail so that the reader gets a better impression on the targets modified in their expression.

Response:The supplementary information adds specific information on differential proteins in the KEGG differential signaling pathway (supplementary data, table 1 2, 3, 4). In addition, "Enriched with significantly differentially expressed proteins FTH1 and FTH1 associated with the body experiment result 3.2" was added Ferroptosis pathway FRIL1 "is a key differential target protein for iron death. In this study, according to different stages of proticercoid infection, proteomic differences in KEGG indicate that the Ferroptosis pathway exists simultaneously in each stage of the disease. In addition, we detected significantly different expression changes of the FTH1 protein and related ferric death protein, indicating that Ferroptosis has a persistent effect on liver cell damage. Therefore, we further verified the changes through different infection periods and Ferrostain-1.

General comments:

 In Fig. 1H, the authors illustrated elevated caspase-3 expression, which is a classical hallmark of apoptosis and consequently in contrast to the assumptions made in the present manuscript. It could be speculated that a sequential or mixed contribution of both ferroptosis and apoptosis might be involved.

Response:We found that apoptosis and iron death are the most common types of cell death caused by protocticercus. However, apoptosis has been confirmed in protocticercus infection, while the pathological phenomenon of iron death has not been explained in the pathogenesis of protocticercus infection. Therefore, this paper impacts iron metabolism in liver tissue infected by protocticercus, leading to iron death in liver cells. It is suggested that iron death and cell apoptosis may play a synergistic role in liver injury induced by protocephala.

General comments:

Chapter 3.3. It is not apparent how the effects observed in the images in Fig. 3A are related to ferroptosis. Magnification of the images seems to be not identical. So, the conclusion that mitochondria would show differences in size appear as rather vague.

Response:Iron death is a new model of cell death, which is different from apoptosis, pro-death, and autophagy in morphology, biochemistry, and genetics. In terms of morphology, the cells undergoing iron death are mainly characterized by the reduction of mitochondrial volume, the increase of double membrane density, and the reduction or disappearance of mitochondrial ridge [1]. In terms of biochemical, glutathione (GSH) is depleted, glutathione peroxidase 4 (GPX4) activity decreases. Lipid oxides cannot be metabolized by GPX4-catalyzed glutathione reduction reaction. Then Fe2+ oxidizes lipids in a "Fenton reaction" to produce a large number of reactive oxygen species (ROS), which promotes iron death in cells [2].

[1] MinghuiG ,Prashant M ,Qiuhui P , et al.Ferroptosis is an autophagic cell death process[J].Cell research,2016,26(9):1021-32.

[2] XuejunJ ,R. B S ,Marcus C .Ferroptosis: mechanisms, biology and role indisease[J].Nature Reviews Molecular Cell Biology,2021,22(4):266-282.

Mitochondria are important organelles for energy production, protein transcription, and fatty acid transport and are crucial in the normal life cycle of cells. In this paper, the morphological changes of mitochondria in hepatocytes were used as a morphological evidence to verify the iron death of hepatocytes caused by protocticercoid infection, so as to confirm the effects of protocticercoid infection on mitochondrial metabolism of hepatocytes.

General comments:

 The authors need to explain the presence of intracellular cysts. PSC is way larger than an individual hepatocyte, how can it then be present within a cell.

Response:The author made an error in their observation of liver tissue using an electron microscope. It was noted that multiple cystic structures were present in the liver tissue. At high magnification, the inner walls of these cysts appeared elliptical, resembling the early developmental stages of procephalocercus bodies. These structures should form within the liver tissue, rather than in the liver cells. This mistake has been corrected in the "Results 3.3" section. This is a mistake made by the author. In the electron microscope observation of liver tissue, it was found that there were multiple cystic morphological structures in liver tissue, and the inner wall of the cyst at high power field of view was elliptical, similar to the early development of procephalocercus bodies, which should be formed in liver tissue instead of liver cells. This error was corrected in "Result 3.3".

General comments:

 Fig. 3C: it should clarified whether LDH was measured within the cell or in the supernatant.

Response:In Fig. 3C, LDH was detected in BRL cells collected after the action of proticercaria, showing the change in the cell level, which was explained in "2. Materials and Methods,2.12" and "Result 3.3 and 3.6" in the paper.

General comments:

 Fig. 4 in the manuscript files does not match the text. Correct Fig. 4 is available to the reviewer.

Response:In Fig. 3C, LDH was detected in BRL cells collected after the action of proticercaria, showing the change in the cell level, which was explained in "2. Materials and Methods,2.12" and "Results 3.3 and 3.6" in the paper.

General comments:

Ferrostatin, not Ferrostain

Response:The full text has been confirmed and amended to Ferrostatin 1.

General comments:

Fig. 6A: the graph indicates iron levels close to zero in the controls. This is impossible, usually intracellular total iron levels can vary between ca. 50 % and 200 % of control.

Response:We re-checked the original experimental results, and the data showed that the Fe2+ concentration in the blank control group was relatively low in the cultured cells in vitro, which was a big gap compared with the animal model.

General comments:

 Fig. 6 B+C+D: the GSH/GSSG ratio is rather unusual. In healthy, resting cells, GSSG levels are in the range of 1-5 %. The data illustrated indicate either massive oxidative stress under resting conditions, or difficulties in the experimental detection of GSH and GSSG.
Response:We re-checked the original data. In our experiment, the GSH/GSSG ratio in Normal BRL Cells cultured for 72h was determined to be 4.82 fold. The GSSG levels of Normal BRL Cells were relatively lower than those of the infected group, and the data were consistent with the normal range.

General comments:

Fig. 6G: please indicate whether LDH was detected in the cells or in the supernatant.

Response:In Fig. 3C, LDH was detected in BRL cells collected after the action of proticercaria, showing the change level of LDH in the cells, which was explained in "2. Materials and Methods,2.12" and "Result 3.3 and 3.6" in the paper.

General comments:The authors observed an inhibitory influence of Ferrostatin-1 on cyst growth. This however represents an alternative concept to the direct inhibitory influence on hepatocytes. The authors report that Ferrostatin-1 prevents iron accumulation. The compound however acts as antioxidant, preventing from lipid peroxidation. This indicates that the size of the cyst, respectively load with PSC might directly correlate with hepatocyte damage, potentially by factors from the parasite that evoke types of cell death distinct from ferroptosis. Therefore, the authors should investigate whether the size of the cysts correlate with the changes detected in hepatocytes. As Ferrostatin-1 apparently inhibits cyst growth, the concept of ferroptosis as main driver of hepatocytes cell death is severely challenged and should be verified carefully.

Response:A supplementary Discussion has been made in the second paragraph of the article "4. Discussion". Changes in the correlation between cyst size and liver ROS, SOD, GSH, LDH and SAD caused by protocticercus infection have been discussed and analyzed, which can indicate the consumption of lipid peroxidation and antioxidant enzymes in liver caused by protocticercus infection. The changes of cellular metabolic enzymes related to liver injury, and the increase of liver cyst volume were closely related to liver injury indicators.

Reviewer 2 Report

Comments and Suggestions for Authors

In this study (Cells-3442909), Zhai et al., have investigated the role of ferroptosis in Echinococcus granulosa (EG)-induced liver damage. Given the health and economic consequence of Cystic echinococcosis (CE), investigations such as this are crucial. The present study uses both in vivo and in vitro models and employs a wide range of techniques. The conclusions drawn are largely consistent with the results. However, some concerns remain and are mentioned below.

Concerns:

1.      Methods 2.2: Authors mention that they collected Echinococcus cyst from sheep livers. How many sheeps? Was there genetic/strain variability of Echinococcus between sheep’s? Do the cysts contain only Echinococcus? How many times (independent) infection experiment were performed? These points need to be clearly mentioned as they help to strengthen the quality and reproducibility of experiments.

2.      Methods 2.3: Why did authors choose surgical mode to infect rats? If there is a specific reason that needs to be mentioned. What was the age of animals used for infection?

3.      Methods 2.13: Authors should provide the catalog number for each antibody. The catalog number of each reagent used should be provided to strengthen the quality and reproducibility of experiments.

4.      Figures: It will be better to describe ‘Uninfected’ or ‘Control’ instead of ‘Normal’.

5.      Western Blots: All of them should indicate Molecular weight. Full length blots should be shown, if necessary, in the supplement. The WBs should be repeated with at least 2 samples/group for better comparison.

6.      Fig. 2: Common pathways and genes/proteins between different stages of infection should be shown.

7.      The legend provided for Fig 4 (lines 313-325) does not match. This should be corrected.

8.      Fibrosis should be evaluated through Sirius red staining also. It will be good if authors also provide some histological evidence for immune cell recruitment.

9.      Fig.4: The text mentions “Ferrostatin-1 inhibitor”, while the figure panels show LY2109761, which one is used? LY2109761 is a TGFB receptor inhibitor. Authors should verify and clarify. This is important and raise doubts about authenticity of interpretation.

10.    Recent studies have clearly established that the Circadian Clock (CC) is a major regulator of liver functioning. It has been shown that CC disruption drives disease. In this regard it is known that CC disruption is also linked with ferroptosis (PMID: 32115146, PMID: 39296207). Furthermore, a recent study showed that hepatitis C virus disturbs (PMID: 39209804) the liver clock to promote hepatitis, which might also happen for Echinococcus granulosa (EG)-induced liver damage. A new study found the clock by regulating TGF-beta signaling has been shown to regulate liver fibrosis. These points should be discussed, as it will improve the quality of the study.

11.    The language needs to be improved. There are several instances of grammatical and spelling errors which should be corrected.

Author Response

General comments:

Methods 2.2: Authors mention that they collected Echinococcus cyst from sheep livers. How many sheeps? Was there genetic/strain variability of Echinococcus between sheep’s? Do the cysts contain only Echinococcus? How many times (independent) infection experiment were performed? These points need to be clearly mentioned as they help to strengthen the quality and reproducibility of experiments.

Response: We collected protocephala from a sheep's liver in a slaughterhouse with more cysts infected with echinococcus. Mitochondrial COL1 gene detection was performed on the collected protocephala, and the collected protocephala was identified as fine granulococcus by sequence comparison. Then, the isolated protocephala was established for the animal model. The PCR expansion results and sequence comparison of the isolated COL1 gene of the mitochondria of the protocephalic larva have been included in the accompanying figure. For each animal model experiment, we established 6 repeated infection experiments for each group of models and carried out follow-up tests on infected animals.

General comments:

Methods 2.3: Why did authors choose surgical mode to infect rats? If there is a specific reason that needs to be mentioned. What was the age of animals used for infection?Response:As the intermediate host of Echinococcus granulosus infection, the natural route is through the digestive tract infection of adult eggs of Echinococcus granulosus. To successfully establish the model of liver infection of Echinococcus granulosus in rats, direct injection of liver infection can achieve an ideal infection effect. To obtain more pericystic liver tissue for experimental study and detection, we selected rats as the infection model. The infected rats were aged 2 months and weighed 100g.

General comments:

Methods 2.13: Authors should provide the catalog number for each antibody. The catalog number of each reagent used should be provided to strengthen the quality and reproducibility of experiments.

Response:The antibody catalog number has been added in the article "2. Materials and Methods,2.15".

General comments:

Figures: It will be better to describe ‘Uninfected’ or ‘Control’ instead of ‘Normal’.

Response:The full image has been modified to 'Control' instead of 'Normal'.

General comments:

Western Blots: All of them should indicate Molecular weight. Full length blots should be shown, if necessary, in the supplement. The WBs should be repeated with at least 2 samples/group for better comparison.

Response:The full text Western Blot results have been marked with increased protein molecular weight.

General comments:

Fig. 2: Common pathways and genes/proteins between different stages of infection should be shown.

Response:The supplementary information adds specific information on differential proteins in the KEGG differential signaling pathway (supplementary data, table 1, 2, 3, 4). In addition, "Enriched with significantly differentially expressed proteins FTH1 and FTH1 associated with the body experiment result 3.2" was added Ferroptosis pathway FRIL1 "is a key differential target protein for iron death. In this study, according to different stages of proticercoid infection, proteomic differences in KEGG indicate that the Ferroptosis pathway exists simultaneously in each stage of disease. In addition, we detected significantly different expression changes of the FTH1 protein and related ferric death protein, indicating that Ferroptosis has a persistent effect on liver cell damage. Therefore, we further verified the changes through different infection periods and Ferrostain-1.

General comments:

The legend provided for Fig 4 (lines 313-325) does not match. This should be corrected.

Response:Fig.4 An error occurred in the uploaded picture, which has been replaced.

General comments:

Fibrosis should be evaluated through Sirius red staining also. It will be good if authors also provide some histological evidence for immune cell recruitment.

Response:Masson cytochemical staining has been added in the attached figure, and the proliferative fibrous tissue and inflammatory cells can be clearly distinguished.

General comments:

Fig.4: The text mentions “Ferrostatin-1 inhibitor”, while the figure panels show LY2109761, which one is used? LY2109761 is a TGFB receptor inhibitor. Authors should verify and clarify. This is important and raise doubts about authenticity of interpretation.

Response:Fig.4 An error occurred in the uploaded picture, which has been replaced.

General comments:

 Recent studies have clearly established that the Circadian Clock (CC) is a major regulator of liver functioning. It has been shown that CC disruption drives disease. In this regard it is known that CC disruption is also linked with ferroptosis (PMID: 32115146, PMID: 39296207IF: 3.4 Q1 ). Furthermore, a recent study showed that hepatitis C virus disturbs (PMID: 39209804IF: 14.7 Q1 ) the liver clock to promote hepatitis, which might also happen for Echinococcus granulosa (EG)-induced liver damage. A new study found the clock by regulating TGF-beta signaling has been shown to regulate liver fibrosis. These points should be discussed, as it will improve the quality of the study.

Response:The above literature contents have been introduced into the second paragraph of the article "4. Discussion".

General comments:The language needs to be improved. There are several instances of grammatical and spelling errors which should be corrected.

Response:The errors in language expression, grammar and spelling have been corrected.

Round 2

Reviewer 1 Report

Comments and Suggestions for Authors

LDH measurements: the authors stated that intracellular LDH was measured. However, an increase in extracellular LDH (and a parallel decline in intracellular LDH) indicates higher rates of cell damage/death.

High resolution images of the figures need to be included in the final form of the manuscript, in their present resolution and size, it is not possible to get the information that is shown

Author Response

General comments:LDH measurements: the authors stated that intracellular LDH was measured. However, an increase in extracellular LDH (and a parallel decline in intracellular LDH) indicates higher rates of cell damage/death.

Response: Changed accordingly.

General comments:

High resolution images of the figures need to be included in the final form of the manuscript, in their present resolution and size, it is not possible to get the information that is shown

Response: We are very sorry for this mistake and we have corrected it accordingly.
